# A General Upper Bound for Unsupervised Domain Adaptation

## Abstract

In this work, we present a novel upper bound of target error to address a major problem faced by many unsupervised domain adaptation methods. Recent studies reveal that a deep neural network can learn transferable features which generalize well to novel tasks. Furthermore, a theory proposed by Ben-David et al. (2010) provides an upper bound for target error when transferring the knowledge, which can be summarized as minimizing the source error and distance between marginal distributions simultaneously. However, common methods based on the theory usually ignore the joint error such that samples from different classes might be mixed together when matching marginal distribution. And in such case, no matter how we minimize the marginal discrepancy, the target error is not bounded due to an increasing joint error. To tackles this problem, we propose a general upper bound taking joint error into account, such that the undesirable case can be properly penalized. In addition, we utilize constrained hypothesis space to further formalize a tighter bound as well as a novel cross margin discrepancy to measure the dissimilarity between hypotheses which alleviates instability during adversarial learning. Extensive empirical evidence shows that our proposal outperforms related approaches in image classification error rates on standard domain adaptation benchmarks.

## 1 Introduction

The advent of deep convolutional neural networks (Krizhevsky et al., 2012) brings visual learning into a new era. However, the performance heavily relies on the abundance of data annotated with ground-truth labels. Since traditional machine learning assumes a model is trained and verified in a fixed distribution (single domain), where generalization performance is guaranteed by VC theory (N. Vapnik, 2000), thus it cannot always be applied to real-world problem directly. Take image classification task as an example, a number of factors, such as the change of light, noise, angle in which the image is pictured, and different types of sensors, can lead to a domain-shift thus harm the performance when predicting on test data.

Therefore, in many practical cases, we wish that a model trained in one or more source domains is also applicable to another domain. As a solution, domain adaptation (DA) aims to transfer the knowledge learned from a source distribution, which is typically fully labeled into a different (but related) target distribution. This work focus on the most challenging case, i.e, unsupervised domain adaptation (UDA), where no target label is available.

Ben-David et al. (2010) suggests that target error can be minimized by bounding the error of a model on the source data, the discrepancy between distributions of the two domains, and a small optimal joint error. Owing to the strong representation power of deep neural nets, many researchers focus on learning domain-invariant features such that the discrepancy of two feature spaces can be minimized. For aligning feature distributions across domains, mainly two strategies have been substantially explored. The first one is bridging the distributions by matching all their statistics (Long et al., 2015; 2017; Pan et al., 2009). The second strategy is using adversarial learning (Goodfellow et al., 2014) to build a minimax game between domain discriminator and feature extractor, where a domain discriminator is trained to distinguish the source from the target while the feature extractor is learned to confuse it simultaneously (Ganin & Lempitsky, 2015; Ganin et al., 2016; Tzeng et al., 2017).

In spite of the remarkable empirical results accomplished by feature distribution matching schemes, they still suffer from a major limitation: the joint distributions of feature spaces and categories are not well aligned across data domains. As is reported in Ganin et al. (2016), such methods fail to generalize in certain closely related source/target pairs, e.g., digit classification adaptation from MNIST to SVHN. One potential reason is when matching marginal distributions of source and target domains, samples from different classes can be mixed together, where the joint error becomes non-negligible since no hypothesis can classify source and target at the same time.

This work aims to address the above problem by incorporating joint error to formalize an optimizable upper bound such that the undesired overlap due to a wrong match can be properly penalized. We evaluate our proposal on several different classification tasks. In some experimental settings, our method outperforms other methods by a large margin. The contributions of this work can be summarized as follows:

· We propose a novel upper bound taking joint error into account and theoretically prove that our proposal can reduce to several other methods under certain simplifications.

· We construct a constrained hypothesis space such that a much tighter bound can be obtained during optimization.

· We adopt a novel measurement, namely cross margin discrepancy, for the dissimilarity of two hypotheses on certain domain to alleviate the instability during adversarial learning and provide reliable performance.

## 2 Related Work

The upper bound proposed by Ben-David et al. (2010) invokes numerous approaches focusing on reducing the gap between source and target domains by learning domain-invariant features, which can be achieved through statistical moment matching. Long et al. (2015; 2017) use maximum mean discrepancy (MMD) to match the hidden representations of certain layers in a deep neural network. Transfer Component Analysis (TCA) (Pan et al., 2011) tries to learn a subspace across domains in a Reproducing Kernel Hilbert Space (RKHS) using MMD that dramatically minimize the distance between domain distributions. Adaptive batch normalization (AdaBN) (Li et al., 2018) modulates the statistics from source to target on batch normalization layers across the network in a parameter-free way.

Another way to learn domain-invariant features is by leveraging generative adversarial network to produce target features that exactly match the source. Ganin & Lempitsky (2015) relax divergence measurement in the upper bound by a worst case which is equivalent to the maximum accuracy that a discriminator can possibly achieve when distinguishing source from target. Tzeng et al. (2017) follow this idea but separate the training procedure into classification stage and adversarial learning stage where an independent feature extractor is used for target. Saito et al. (2017b) explore a tighter bound by explicitly utilizing task-specific classifiers as discriminators such that features nearby the support of source samples will be favored by extractor. Zhang et al. (2019) introduce margin disparity discrepancy, a novel measurement with rigorous generalization bounds, tailored to the distribution comparison with the asymmetric margin loss to bridge the gap between theory and algorithm. Methods perform distribution alignment on pixel-level in raw input, which is known as image-to-image translation, are also proposed (Liu & Tuzel, 2016; Bousmalis et al., 2017; Sankaranarayanan et al., 2017; Shrivastava et al., 2016; Hoffman et al., 2018; Murez et al., 2017).

Distribution matching may not only bring the source and target domains closer, but also mix samples with different class labels together. Therefore, Saito et al. (2017a); Sener et al. (2016); Zhang et al. (2018) aim to use pseudo-labels to learn target discriminative representations encouraging a low-density separation between classes in the target domain (Lee, 2013). However, this usually requires auxiliary data-dependent hyper-parameter to set a threshold for a reliable prediction. Long et al. (2018) present conditional adversarial domain adaptation, a principled framework that conditions the adversarial adaptation models on discriminative information conveyed in the classifier predictions, where the back-propagation of training objective is highly dependent on pseudo-labels.

# 3 Proposed Method

## 3.1 A General Upper Bound

We consider the unsupervised domain adaptation as a binary classification task (our proposal holds for multi-class case) where the learning algorithm has access to a set of $n$ labeled points $\{(x_s^i, y_s^i) \in (X \times Y)\}_{i=1}^n$ sampled i.i.d. from the source domain $S$ and a set of $m$ unlabeled points $\{(x_t^i) \in X\}_{i=1}^m$ sampled i.i.d. from the target domain $T$. Let $f_S : X \to \{0,1\}$ and $f_T : X \to \{0,1\}$ be the optimal labeling functions on the source and target domains, respectively. Let $\epsilon$ (usually 0-1 loss) denotes a distance metric between two functions over a distribution that satisfies symmetry and triangle inequality. As a commonly used notation, the source risk of hypothesis $h : X \to \{0,1\}$ is the error w.r.t. the true labeling function $f_S$ under domain $S$, i.e., $\epsilon_S(h) := \epsilon_S(h, f_S)$. Similarly, we use $\epsilon_T(h)$ to represent the risk of the target domain. With these notations, the following bound holds:

$$
\begin{aligned}
\epsilon_T(h) &= \epsilon_T(h, f_T) \\
&= \epsilon_T(h, f_T) - \epsilon_T(h, f_S) + \epsilon_T(h, f_S) + \epsilon_S(h, f_S) - \epsilon_S(h, f_S) + \epsilon_S(h, f_T) - \epsilon_S(h, f_T) \\
&= \epsilon_S(h, f_S) + (\epsilon_T(h, f_T) - \epsilon_T(h, f_S)) + (\epsilon_S(h, f_T) - \epsilon_S(h, f_S)) + \epsilon_T(h, f_S) - \epsilon_S(h, f_T) \\
&\leq \epsilon_S(h) + \epsilon_T(f_S, f_T) + \epsilon_S(f_S, f_T) + \epsilon_T(h, f_S) - \epsilon_S(h, f_T) \\
&= \epsilon_S(h) + C_{S,T}(f_S, f_T, h)
\end{aligned}
\tag{1}
$$

For simplicity, we use $C_{S,T}(f_S, f_T, h)$ to denote $\epsilon_T(f_S, f_T) + \epsilon_S(f_S, f_T) + \epsilon_T(h, f_S) - \epsilon_S(h, f_T)$. The above upper bound is minimized when $h = f_S$, and it is equivalent to $\epsilon_T(f_S, f_T)$ owing to the triangle inequality:

$$
\begin{aligned}
\epsilon_S(h) + \epsilon_S(f_S, f_T) = \epsilon_S(h, f_S) + \epsilon_S(f_S, f_T) \\
\geq \epsilon_S(h, f_T)
\end{aligned}
\tag{2}
$$

Furthermore, we demonstrate in such case, our proposal is equivalent to an upper bound of optimal joint error $\lambda$ because:

$$
\begin{aligned}
\epsilon_T(f_S, f_T) &= \epsilon_T(f_S, f_T) + \epsilon_S(f_S, f_S) \\
&= \epsilon_T(f_S) + \epsilon_S(f_S) \\
&\geq \min_h(\epsilon_T(h) + \epsilon_S(h)) = \lambda
\end{aligned}
\tag{3}
$$

Fig. 1b illustrates a case where common methods fail to penalize the undesirable situation when samples from different classes are mixed together during distribution matching, while our proposal is capable to do so (for simplicity we assume $f_S$ takes a specific form, then $\epsilon_T(f_S, f_T)$ measures the overlapping area 2 and 5, which is equivalent to the optimal joint error $\lambda$).

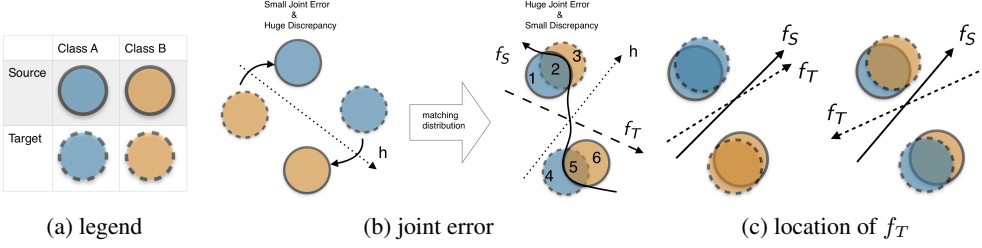

(a) legend          (b) joint error          (c) location of $f_T$

Figure 1: (a) Legend used in entire paper. (b) Joint error (area 2 and 5) is penalized such that extractor must try to separate the overlap. (c) $f_T$ does not necessarily classify source samples.

## 3.2 Hypothesis Space Constraint

Since optimal labeling functions $f_S, f_T$ are not available during training, we shall further relax the upper bound by taking supreme w.r.t $f_S, f_T$ within a hypothesis space $H$:

$$\begin{aligned} \epsilon_T(h) &\leq \epsilon_S(h) + C_{S,T}(f_S, f_T, h) \\ &\leq \epsilon_S(h) + \sup_{f_1, f_2 \in H} C_{S,T}(f_1, f_2, h) \end{aligned} \tag{4}$$

Then minimizing target risk $\epsilon_T(h)$ becomes optimizing a minimax game and since the max-player taking two parameters $f_1, f_2$ is too strong, we introduce a feature extractor $g$ to make the min-player stronger. Applying $g$ to the source and target distributions, the overall optimization problem can be written as:

$$\min_{g,h}(\epsilon_{g(S)}(h) + \max_{f_1, f_2 \in H} C_{g(S), g(T)}(f_1, f_2, h)) \tag{5}$$

However, if we leave $H$ unconstrained, the supreme term can be arbitrary large. In order to obtain a tight bound, we need to restrict the size of hypothesis space as well as maintain the upper bound. For $f_S \in H_1 \leq H$ and $f_T \in H_2 \leq H$, the following holds:

$$C_{g(S), g(T)}(f_S, f_T, h) \leq \sup_{f_1 \in H_1, f_2 \in H_2} C_{g(S), g(T)}(f_1, f_2, h) \leq \sup_{f_1, f_2 \in H} C_{g(S), g(T)}(f_1, f_2, h) \tag{6}$$

The constrained subspace for $H_1$ is trivial as according to its definition, $f_S$ must belong to the space consisting of all classifiers for source domain, namely $H_{sc}$. However, the constrained subspace for $H_2$ is a little problematic since we have no access to the true labels of the target domain, thus it is hard to locate $f_T$. Therefore, the only thing we can do is to construct a hypothesis space for $H_2$ that most likely contains $f_T$. As is illustrated in Fig. 1c, when matching distributions of source and target domain, if the ideal case is achieved where the conditional distributions of source and target are perfectly aligned, then it is fare to assume $f_T \in H_{sc}$. However, if the worst case is reached where samples from different class are mixed together, then we tend to believe $f_T \notin H_{sc}$. Considering this, we present two proposals in the following sections based on different constraints.

### 3.2.1 Original Proposal

We assume $H_2$ is a space where the hypothesis can classify the samples from the source domain with an accuracy of $\gamma \in [0, 1]$, namely $H_{sc}^\gamma$, such that we can avoid the worst case by choosing a small value for the hyper-parameter $\gamma$ when a huge domain shift exists. In practice, it is difficult to actually build such a space and sample from it due to a huge computational cost. Instead, we use a weighted source risk to constrain the behavior of $f_2$ as an approximation to the sample from $H_{sc}^\gamma$, which leads to the final training objective:

$$\begin{cases} \min_{g,h}(\epsilon_{g(S)}(h) + \max_{f_1, f_2}(\epsilon_{g(T)}(f_1, f_2) + \epsilon_{g(S)}(f_1, f_2) + \epsilon_{g(T)}(h, f_1) - \epsilon_{g(S)}(h, f_2))) \\ s.t. \quad \min_{g, f_1, f_2}(\epsilon_{g(S)}(f_1) + \gamma \epsilon_{g(S)}(f_2)) \end{cases} \tag{7}$$

### 3.2.2 Alternative Proposal

Firstly, we build a space consisting of all classifiers for approximate target domain $\{(x_t^i, h(x_t^i)) \in X \times Y\}_{i=1}^m$ based on pseudo labels which can be obtained by the prediction of $h$ during training procedure, namely $H_{\tilde{t}c}$. Here, we assume $H_2$ is an intersection between two hypothesis spaces , i.e. $H_{sc}^\eta \cap H_{\tilde{t}c}^{1-\eta}$ where the hypothesis can classify the samples from source domain with an accuracy of $\eta \in [0, 1]$ and classify the samples from approximate target domain with an accuracy of $1 - \eta$. Given enough reliable pseudo labels, we can be confident about $f_T \in H_2$. Analogously, the training objective is given by:

$$\begin{cases} \min_{g,h}(\epsilon_{g(S)}(h) + \max_{f_1,f_2}(\epsilon_{g(T)}(f_1,f_2) + \epsilon_{g(S)}(f_1,f_2) + \epsilon_{g(T)}(h,f_1) - \epsilon_{g(S)}(h,f_2))) \\ s.t. \quad \min_{g,f_1,f_2}(\epsilon_{g(S)}(f_1) + \eta\epsilon_{g(S)}(f_2) + (1-\eta)\tilde{\epsilon}_{g(T)}(f_2)) \end{cases}$$

(8)

### 3.2.3 Intuition

The reason we make such an assumption for $H_2$ can be intuitively explained by Fig. 2. If $H_2 = H_{sc}$, then $f_2$ must perfectly classify the source samples, and it is possible that $f_2$ does not pass through some target samples (shadow are in 2a), especially when two domains differ a lot. In such case, the feature extractor can move those samples into either side of the decision boundary to reduce the training objective (shadow area) which is not a desired behavior. With an appropriate constraint (2b), as for the extractor, the only way to reduce the objective (shadow area) is to move those samples (orange) inside of $f_2$.

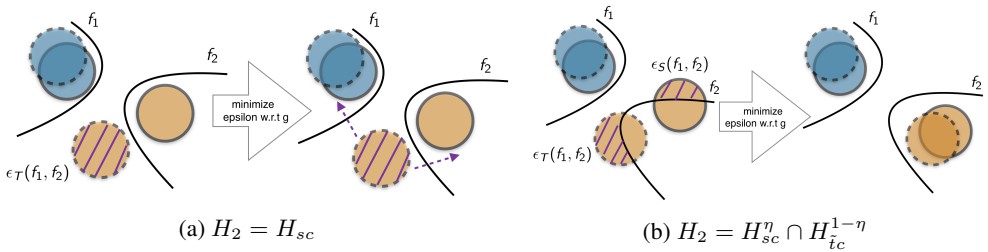

(a) $H_2 = H_{sc}$          (b) $H_2 = H_{sc}^{\eta} \cap H_{\tilde{t}c}^{1-\eta}$

Figure 2: (a) Failure due to improper constraint. (b) Proper constraint helps for huge domain shift.

## 3.3 Cross Margin Discrepancy

Following the above notations, we consider a score function $s(x,y)$ for multi-class classification where the output indicates the confidence of the prediction on class $y$. Thus an induced labeling function named $l_s$ from $X \to Y$ is given by:

$$l_s : x \to \arg\max_{y \in Y} s(x,y)$$

(9)

As a well-established theory, the margin between data points and the classification surface plays a significant role in achieving strong generalization performance. In order to quantify $\epsilon$ into differentiable measurement as a surrogate of 0-1 loss, we introduce the margin theory developed by Koltchinskii & Panchenko (2002), where a typical form of margin loss can be interpreted as:

$$\mathbb{E}_{(x,y)\in D}[\max(0, 1 + \max_{y'\neq y} s(x,y') - s(x,y))]$$

(10)

We aim to utilize this concept to further improve the reliability of our proposed method by leveraging this margin loss to define a novel measurement of the discrepancy between two hypotheses $f_1, f_2$ (e.g. softmax) over a distribution $D$, namely cross margin discrepancy:

$$\epsilon_D(f_1, f_2) = \mathbb{E}_{x\in D}[d(f_1, f_2, x)]$$

(11)

Before further discussion, we firstly construct two distributions $D_{f_1}, D_{f_2}$ induced by $f_1, f_2$ respectively, where $D_{f_1} = \{(x, l_{f_1}(x))|x \sim D\}$ and $D_{f_2} = \{(x, l_{f_2}(x))|x \sim D\}$. Then we consider the case where two hypotheses $f_1$ and $f_2$ disagree, i.e. $y_1 = l_{f_1}(x) \neq l_{f_2}(x) = y_2$, and the primitive loss is defined as:

$$\begin{aligned} d(f_1, f_2, x) &= \log f_1(x, y_1) - \log f_2(x, y_1) + \log f_2(x, y_2) - \log f_1(x, y_2) \\ &= \log f_1(x, y_1) - \log f_1(x, y_2) + \log f_2(x, y_2) - \log f_2(x, y_1) \end{aligned}$$

(12)

Then the cross margin discrepancy can be viewed as:

$$\epsilon_D(f_1, f_2) = \mathbb{E}_{(x,y)\in D_{f_2}}[\max_{y'\neq y}\log f_1(x,y')-\log f_1(x,y)]+\mathbb{E}_{(x,y)\in D_{f_1}}[\max_{y'\neq y}\log f_2(x,y')-\log f_2(x,y)]$$
(13)

which is a sum of the margin loss for $f_1$ on $D_{f_2}$ and the margin loss for $f_2$ on $D_{f_1}$, if we use the logarithm of $\mathrm{softmax}$ as the score function.

Thanks to the trick introduced by Goodfellow et al. (2014) to mitigate the burden of exploding or vanishing gradients when performing adversarial learning, we further define a dual form as:

$$d(f_1, f_2, x) = \log f_1(x, y_1) + \log(1 - f_1(x, y_2)) + \log f_2(x, y_2) + \log(1 - f_2(x, y_1))$$
(14)

This dual loss resembles the objective of the generative adversarial network, where two hypotheses try to increase the probability of their own prediction and simultaneously decrease the probability of their opponents; whereas the feature extractor is trained to increase the probability of their opponents, such that the discrepancy can be minimized without unnecessary oscillation. However, a big difference here is when training extractor, GANs usually maximize an alternative term $\log f_1(x, y_2) + \log f_2(x, y_1)$ instead of directly minimizing $\log(1 - f_1(x, y_2)) + \log(1 - f_2(x, y_1))$ since the original term is close to zero if the discriminator achieves optimum. In our case, the hypothesis can hardly beat the extractor thus the original form can be more smoothly optimized.

During the training procedure, the two hypotheses will eventually agree on some points ($l_{f_1}(x) = l_{f_2}(x) = y$) such that we need to define a new form of discrepancy measurement. Analogously, the primitive loss and its dual form are given by:

$$d(f_1, f_2, x) = \log\max(f_1(x, y), f_2(x, y)) - \log\min(f_1(x, y), f_2(x, y))$$
(15)
$$d(f_1, f_2, x) = \log\max(f_1(x, y), f_2(x, y)) + \log\max(1 - f_1(x, y), 1 - f_2(x, y))$$
(16)

Another reason why we propose such a discrepancy measurement is that it helps alleviate instability for adversarial learning. As is illustrated in Fig. 3b, during optimization of a minimax game, when two hypotheses try to maximize the discrepancy (shadow area), if one moves too fast around the decision boundary such that the discrepancy is actually maximized w.r.t some samples, then these samples can be aligned on either side to decrease the discrepancy by tuning the feature extractor, which is not a desired behavior. From Fig. 3a, we can see that our proposed cross margin discrepancy is flat for the points around original, i.e. the gradient w.r.t those points nearby the decision boundary will be relatively small, which helps to prevent such failure.

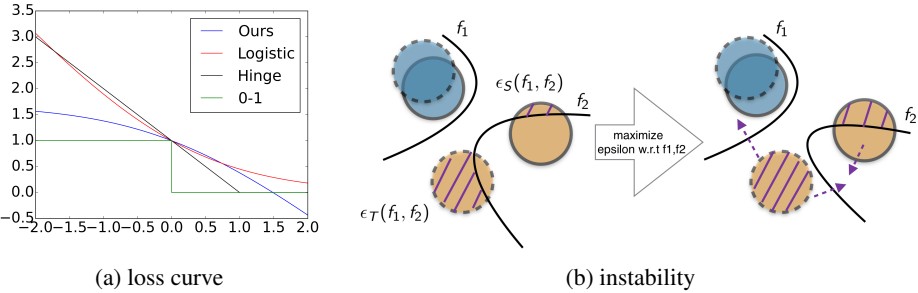

(a) loss curve                      (b) instability

Figure 3: (a) Comparisons for binary classification case. (b) Failure due to steep gradient nearby the decision boundary.

## 3.4 Comparisons with Other Methods

### 3.4.1 Margin Disparity Discrepancy

Zhang et al. (2019) propose a novel margin-aware generalization bound based on scoring functions and a new divergence MDD. The training objective used in MDD can be alternatively interpreted as (here $\epsilon(h, f)$ denotes the margin disparity):

$$\min_{g,h}(\epsilon_{g(S)}(h) + \max_f(\epsilon_{g(T)}(f, h) - \epsilon_{g(S)}(f, h)))$$
(17)

Recall Eq.7, if we set $f_1 = f_2 = f$ and free the constraint of $f$ to any $f \in H$, our proposal degrades exactly to MDD. As is discussed above, when matching distribution, if and only if the ideal case is achieved, where the conditional distributions of induced feature spaces for source and target perfectly match (which is not always possible), can we assume two optimal labeling functions $f_S, f_T$ to be identical. Besides, an unconstrained hypothesis space for $f$ is definitely not helpful to construct a tight bound.

### 3.4.2 Maximum Classifier Discrepancy

Saito et al. (2017b) propose two task-specific classifiers $f_1, f_2$ that are used to separate the decision boundary on source domain, such that the extractor is encouraged to produce features nearby the support of the source samples. The objective used in MCD can be alternatively interpreted as (here $\epsilon(f_1, f_2)$ is quantified by $L_1$):

$$\begin{cases} \min_{g,f_1}(\epsilon_{g(S)}(f_1) + \max_{f_1,f_2}(\epsilon_{g(T)}(f_1, f_2))) \\ s.t. \quad \min_{g,f_1,f_2}(\epsilon_{g(S)}(f_1) + \epsilon_{g(S)}(f_2)) \end{cases} \tag{18}$$

Again, recall Eq.7, if we set $\gamma = 1$ and $h = f_1$, MCD is equivalent to our proposal. As is proved in section 3.1, the upper bound is optimized when $h = f_S$. However, it no longer holds since the upper bound is relaxed by taking supreme to form an optimizabel objective, i.e. setting $h = f_1$ does not necessarily minimize the objective. Besides, as we discuss above, a fixed $\gamma = 1$, i.e $H_2 = H_{sc}$ lacks generality since we have no idea about where $f_T$ might be, such that it is not likely to be applicable to those cases where a huge domain shift exists.

## 4 Evaluation

### 4.1 Experiment on Digit Dataset

In this experiment, our proposal is assessed in four types of adaptation scenarios by adopting commonly used digits datasets (Fig. 6 in Appendix),i.e. MNIST (LeCun et al., 1998), Street View House Numbers (SVHN) (Netzer et al., 2011), and USPS (Hull, 1994) such that the result could be easily compared with other popular methods. All experiments are performed in an unsupervised fashion without any kinds of data augmentation.

### 4.1.1 Experimental Setting

Details are omitted due to the limit of space (see A.1).

### 4.1.2 Result

We report the accuracy of different methods in Tab. 1. Our proposal outperforms the competitors in almost all settings except a single result compared with GPDA (Kim et al., 2019). However, their solution requires sampling that increases data size and is equivalent to adding Gaussian noise to the last layer of a classifier, which is considered as a type of augmentation. Our success partially owes to combining the upper bound with the joint error, especially when optimal label functions differ from each other (e.g. MNIST →SVHN). Moreover, as most scenarios are relatively easy for adaptation thus we can be more confident about the hypothesis space constraint owing to reliable pseudo-labels, which leads to a tighter bond during optimization. The results demonstrate our proposal can improve generalization performance by adopting both of these advantages.

Fig. 4a shows that our original proposal is quite sensitive to the hyper-parameter $\gamma$. In short, setting $\gamma = 1$ here yields the best performance in most situations, since $f_S, f_T$ can be quite close after aligning distributions, especially in these easily adapted scenarios. However, in MNIST $\rightarrow$ SVHN, setting $\gamma = 0.1$ gives the optimum which means that $f_S, f_T$ are so far away due to a huge domain

Table 1: Results of the adaptation experiment on the digits datasets (note that † means a different setting which use labeled target samples for validation; MNIST* and USPS* denote the whole training set; ours: original proposal and $L_1$ ($\gamma = 1$); ours*: original proposal and cross margin discrepancy ($\gamma = 1$); ours⋆: alternative proposal and cross margin discrepancy ($\eta = 0$)).

| METHOD | SVHN to MNIST | MNIST to SVHN | MNIST to USPS | MNIST* to USPS* | USPS to MNIST |
|---|---|---|---|---|---|
| Source Only | 67.1 | 21.3 | 76.7 | 79.7 | 63.4 |
| MDD†(Long et al., 2015) | 71.1 | - | - | 81.1 | - |
| DANN†(Ganin et al., 2016) | 71.1 | 25.1 | 77.3 | 85.1 | 73.2 |
| DRCN(Ghifary et al., 2016) | 82.0±0.1 | 40.1±0.1 | 91.8±0.1 | - | 73.7±0.1 |
| ADDA(Tzeng et al., 2017) | 76.0±1.8 | - | 89.4±0.2 | - | 90.1±0.8 |
| MCD(Saito et al., 2017b) | 96.2±0.4 | 11.2±1.1 | 94.2±0.7 | 96.5±0.3 | 94.1±0.3 |
| GPDA(Kim et al., 2019) | 98.2±0.1 | - | 96.5±0.2 | 98.1±0.1 | 96.4±0.1 |
| ours | 96.8±0.2 | 30.4±1.5 | 94.5±0.3 | 96.8±0.3 | 95.2±0.2 |
| ours* | 97.5±0.2 | 31.5±1.8 | 95.3±0.3 | 97.2±0.2 | 95.6±0.2 |
| ours⋆ | 98.6±0.1 | 50.3±1.3 | 96.8±0.2 | 97.9±0.1 | 96.9±0.1 |

shift that no extractor is capable of introducing an identical conditional distribution in feature space. The improvement is not that much, but at least we outperform the directly comparable MCD and show the importance of hypothesis space constraint. Furthermore, Fig. 4d empirically proves simply minimizing the discrepancy between the marginal distribution does not necessarily lead to a reliable adaptation, which demonstrates the importance of joint error. In addition, Fig. 4b,Fig. 4c show the superiority of the cross margin discrepancy which accelerates the convergence and provides a slightly better result.

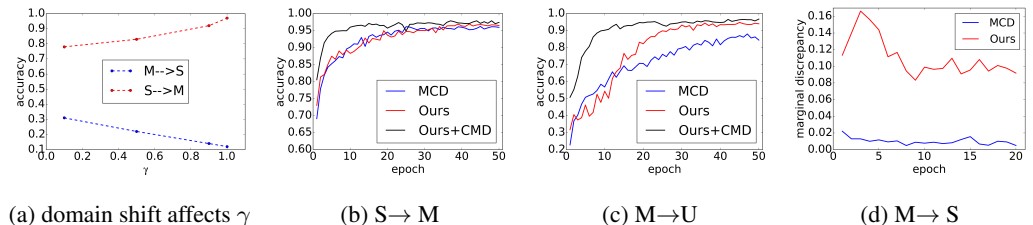

(a) domain shift affects $\gamma$     (b) S→ M     (c) M→U     (d) M→ S

Figure 4: (a) Sensitivity w.r.t. $\gamma$. (b)-(c) Comparisons for convergence rate. (d) Comparisons for marginal discrepancy.

## 4.2 Experiment on VisDA Dataset

We further evaluate our method on object classification. The VisDA dataset (Peng et al., 2017) is used here, which is designed for 12-class adaptation task from synthetic object to real object images. Source domain contains 152,397 synthetic images (Fig. 7a in Appendix), which are generated by rendering 3D CAD models. Data of the target domain is collected from MSCOCO (Lin et al., 2014) consisting of 55,388 real images (Fig. 7b in Appendix). Since the 3D models are generated without the background and color diversity, the synthetic domain is quite different from the real domain, which makes it a much more difficult problem than digits adaptation. Again, this experiment is performed in unsupervised fashion and no data augmentation technique excluding horizontal flipping is allowed.

### 4.2.1 Experimental Setting

Details are omitted due to the limit of space (see A.2).

Table 2: The accuracy of ResNet-101 model fine-tuned on the VisDA dataset within 10 epoch updates (ours: original proposal and $L_1$ ($\gamma = 1$); ours*: original proposal and cross margin discrepancy ($\gamma = 1$); ours$^\star$: alternative proposal and cross margin discrepancy ($\eta = 0.9$)).

| METHOD | plane | bcycl | bus | car | horse | knife | mcycl | person | plant | sktbrd | train | truck | avg |
|---|---|---|---|---|---|---|---|---|---|---|---|---|---|
| Source Only | 55.1 | 53.3 | 61.9 | 59.1 | 80.6 | 17.9 | 79.7 | 31.2 | 81.0 | 26.5 | 73.5 | 8.5 | 52.4 |
| MDD(Long et al., 2015) | 87.1 | 63.0 | 76.5 | 42.0 | 90.3 | 42.9 | 85.9 | 53.1 | 49.7 | 36.3 | 85.8 | 20.7 | 61.1 |
| DANN(Ganin et al., 2016) | 81.9 | 77.7 | 82.8 | 44.3 | 81.2 | 29.5 | 65.1 | 28.6 | 51.9 | 54.6 | 82.8 | 7.8 | 57.4 |
| MCD(Saito et al., 2017b) | 87.0 | 60.9 | 83.7 | 64.0 | 88.9 | 79.6 | 84.7 | 76.9 | 88.6 | 40.3 | 83.0 | 25.8 | 71.9 |
| GPDA(Kim et al., 2019) | 83.0 | 74.3 | 80.4 | 66.0 | 87.6 | 75.3 | 83.8 | 73.1 | 90.1 | 57.3 | 80.2 | 37.9 | 73.3 |
| ours | 86.3 | 82.7 | 83.7 | 68.7 | 87.9 | 72.7 | 85.4 | 61.5 | 87.3 | 55.5 | 75.2 | 34.1 | 73.4 |
| ours* | 88.4 | 83.3 | 74.8 | 78.0 | 88.1 | 43.2 | 88.2 | 68.9 | 87.6 | 65.5 | 92.6 | 58.5 | 76.4 |
| ours$^\star$ | 91.5 | 80.3 | 75.5 | 66.1 | 91.4 | 87.6 | 85.2 | 78.7 | 91.2 | 77.2 | 82.8 | 48.9 | 79.7 |

### 4.2.2 Result

We report the accuracy of different methods in Tab. 2, and find that our proposal outperforms the competitors in all settings. The image structure of this dataset is more complex than that of digits, yet our method provides reliable performance even under such a challenging condition. Another key observation is that some competing methods (e.g., DANN, MCD), which can be categorized as distribution matching based on adversarial learning, perform worse than MDD which simply matches statistics, in classes such as plane and horse, while our methods perform better across all classes, which clearly demonstrates the importance of taking the joint error into account.

As for the original proposal (Fig. 5c), performance drops when relaxing the constraint which actually confuses us. Because we expect an improvement here since it is unbelievable that $f_S$, $f_T$ eventually lie in a similar space judging from the relatively low prediction accuracy. As for the alternative proposal (Fig. 5d), we test the adaptation performance for different $\eta$ and the prediction accuracy drastically drops when $\eta$ goes beyond 0.2. One possible cause is that $f_2$ and $h$ might almost agree on target domain, such that the prediction of $h$ could not provide more accurate information for the target domain without introducing noisy pseudo labels. Fig. 5a, Fig. 5b again demonstrate the superiority of cross margin discrepancy and the importance of joint error.

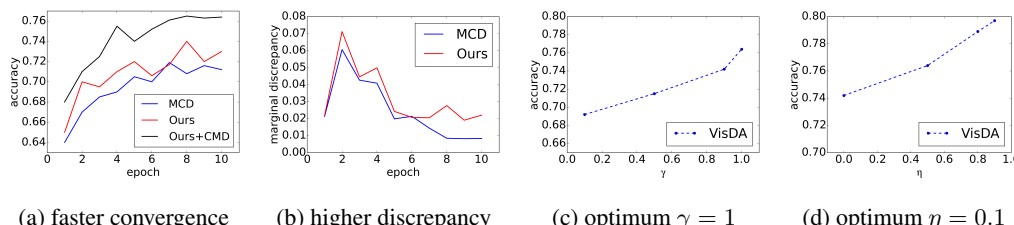

(a) faster convergence     (b) higher discrepancy     (c) optimum $\gamma = 1$     (d) optimum $\eta = 0.1$

Figure 5: (a) Comparisons for convergence rate. (b) Comparisons for marginal discrepancy. (c) Sensitivity w.r.t. $\gamma$. (d) Sensitivity w.r.t. $\eta$.

## 5 Conclusion

In this work, we propose a general upper bound that takes the joint error into account. Then we further pursue a tighter bound with reasonable constraint on the hypothesis space. Additionally, we adopt a novel cross domain discrepancy for dissimilarity measurement which alleviates the instability during adversarial learning. Extensive empirical evidence shows that learning an invariant representation is not enough to guarantee a good generalization in the target domain, as the joint error matters especially when the domain shift is huge. We believe our results take an important step towards understanding unsupervised domain adaptation, and also stimulate future work on the design of stronger adaptation algorithms that manage to align conditional distributions without using pseudo-labels from the target domain.

**Acknowledgments**

This work was partially supported by JST CREST Grant Number JPMJCR1403, and partially supported by JSPS KAKENHI Grant Number JP19H01115. We would like to thank Yusuke Mukuta, Toshihiko Matsuura, Akihiro Nakamura, Wataru Kawai and Hao-Wei Yeh for their thoughtful comments and suggestions.

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

# A    Appendix

## A.1    Experimental Setting for Digits Dataset

Throughout the experiment, we employ the CNN architecture used in Saito et al. (2017b), where batch normalization is applied to each layer and a $0.5$ rate of dropout is used between fully connected layers. Besides, spectral normalization (Miyato et al., 2018) is deployed for the classifiers in all of the following experiments to stabilize adversarial learning. Adam (Kingma & Ba, 2014) is used for optimization with a mini-batch size of 128 and a learning rate of $10^{-4}$. As for the original proposal, we verify the sensitivity of our model on various value for $\gamma = \{0.1, 0.5, 0.9, 1.0\}$. As for the alternative proposal, setting $\eta = 0$ usually provides a reliable performance and the adaptation result changes subtly for different $\eta$.

**SVHN $\leftrightarrow$ MNIST** We firstly examine the adaptation from SVHN (Fig. 6a) to MNIST (Fig. 6b). We use the standard training set and testing set for both source and target domains. The feature extractor contains three $5 \times 5$ convolutional layers with stride two $3 \times 3$ max pooling placed after the first two convolutional layers and a single fully-connected layer. For classifiers, we use 2-layer fully-connected networks.

**MNIST $\leftrightarrow$ USPS** As for the adaptation between MNIST and USPS (Fig. 6c), we follow the training protocol established in Long et al. (2013) by sampling 2000 images from MNIST and 1800 from USPS. As for the test samples, the standard version is used for both source and target domains. The feature generator contains two $5 \times 5$ convolutional layers with stride two $2 \times 2$ max pooling placed after each convolutional layer and a single fully-connected layer. For classifiers, we use 2-layer fully-connected networks.

## A.2    Experimental Setting for VisDA Dataset

Following the protocol in Saito et al. (2017b), we evaluate our method by fine-tuning a ResNet-101 (He et al., 2015) model pretrained on ImageNet (Deng et al., 2009). The model except the last layer combined with a single-layer bottleneck is used as feature extractor and a randomly initialized 2-layer fully-connected network is used as a classifier, where batch normalization is applied to each

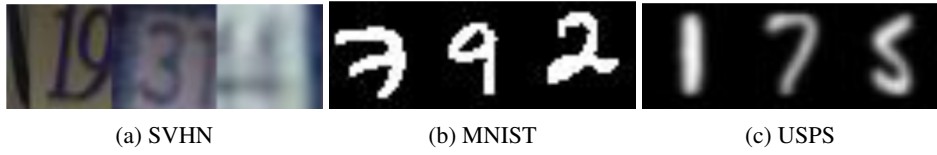

(a) SVHN        (b) MNIST        (c) USPS

Figure 6: Random samples from each dataset.

layer and a $0.5$ rate of dropout is conducted. Nesterov accelerated gradient is used for optimization with a mini-batch size of 32 and an initial learning rate of $10^{-3}$ which decays exponentially. As for the hyper-parameter, we test for $\gamma = \{0.1, 0.5, 0.9, 1\}$ and $\eta = \{0, 0.5, 0.8, 0.9\}$. For a direct comparison, we report the accuracy after 10 epochs.

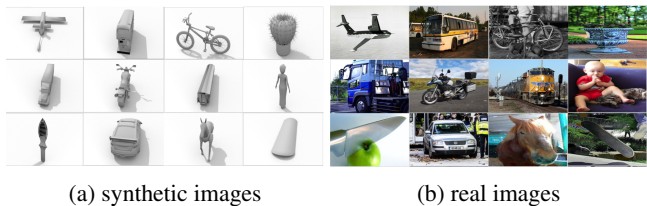

(a) synthetic images        (b) real images

Figure 7: (a) Samples from source domain. (b) Samples from target domain.

## A.3    Additional Experiment on Office-Home Dataset

Office-Home (Venkateswara et al., 2017) is a complex dataset (Fig. 8) containing 15,500 images from four significantly different domains: Art (paintings, sketches and/or artistic depictions), Clipart (clip art images), Product (images without background), and Real-world (regular images captured with a camera). In this experiment, following the protocol from Zhang et al. (2019), we evaluate our method by fine-tuning a ResNet-50 (He et al., 2015) model pretrained on ImageNet (Deng et al., 2009). The model except the last layer combined with a single-layer bottleneck is used as feature extractor and a randomly initialized 2-layer fully-connected network with width 1024 is used as a classifier, where batch normalization is applied to each layer and a $0.5$ rate of dropout is conducted. For optimization, we use the SGD with the Nesterov momentum term fixed to 0.9, where the batch size is 32 and learning rate is adjusted according to Ganin et al. (2016).

From Tab. 3, we can see the adaptation accuracy of the source-only method is rather low, which means a huge domain shift is quite likely to exist. In such case, simply minimizing the discrepancy between source and target might not work as the joint error can be increased when aligning distributions, thus the assumption of the basic theory (Ben-David et al., 2010) does not hold anymore. On the other hand, our proposal incorporates the joint error into the target error upper bound which can boost the performance especially when there is a large domain shift.

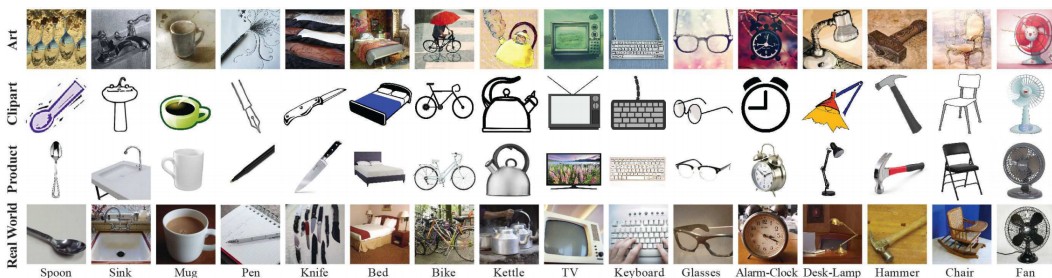

Figure 8: Sample images from the Office-Home dataset (Venkateswara et al., 2017).

Table 3: The accuracy of ResNet-50 model fine-tuned on the Office-Home dataset ( ours*: alternative proposal and cross margin discrepancy ($\eta = 0.9$)).

| METHOD | Ar→Cl | Ar→Pr | Ar→Rw | Cl→Ar | Cl→Pr | Cl→Rw | Pr→Ar | Pr→Cl | Pr→Rw | Rw→Ar | Rw→Cl | Rw→Pr | Avg |
|---|---|---|---|---|---|---|---|---|---|---|---|---|---|
| Source Only | 34.9 | 50.0 | 58.0 | 37.4 | 41.9 | 46.2 | 38.5 | 31.2 | 60.4 | 53.9 | 41.2 | 59.9 | 46.1 |
| DANN(Ganin et al., 2016) | 45.6 | 59.3 | 70.1 | 47.0 | 58.5 | 60.9 | 46.1 | 43.7 | 68.5 | 63.2 | 51.8 | 76.8 | 57.6 |
| CDAN(Long et al., 2018) | 50.7 | 70.6 | 76.0 | 57.6 | 70.0 | 70.0 | 57.4 | 50.9 | 77.3 | 70.9 | 56.7 | 81.6 | 65.8 |
| MDD(Zhang et al., 2019) | 54.9 | 73.7 | 77.8 | 60.0 | 71.4 | 71.8 | 61.2 | 53.6 | 78.1 | 72.5 | 60.2 | 82.3 | 68.1 |
| ours* | 56.9 | 75.8 | 79.3 | 65.4 | 70.7 | 70.8 | 63.5 | 55.7 | 79.4 | 73.5 | 59.3 | 82.1 | 69.4 |

Table 4: The accuracy of ResNet-50 model fine-tuned on the Office-31 dataset ( ours*: alternative proposal and cross margin discrepancy ($\eta = 0.9$)).

| METHOD | A→W | D→W | W→D | A→D | D→A | W→A | Avg |
|---|---|---|---|---|---|---|---|
| Source Only | 68.4±0.2 | 96.7±0.1 | 99.3±0.1 | 68.9±0.2 | 62.5±0.3 | 60.7±0.3 | 76.1 |
| DANN(Ganin et al., 2016) | 82.0±0.4 | 96.9±0.2 | 99.1±0.1 | 79.7±0.4 | 68.2±0.4 | 67.4±0.5 | 82.2 |
| ADDA(Tzeng et al., 2017) | 86.2±0.5 | 96.2±0.3 | 98.4±0.3 | 77.8±0.3 | 69.5±0.4 | 68.9±0.5 | 82.9 |
| MCD(Saito et al., 2017b) | 88.6±0.2 | 98.5±0.1 | 100.0±.0 | 92.2±0.2 | 69.5±0.1 | 69.7±0.3 | 86.5 |
| CDAN(Long et al., 2018) | 94.1±0.1 | 98.6±0.1 | 100.0±.0 | 92.9±0.2 | 71.0±0.3 | 69.3±0.3 | 87.7 |
| MDD(Zhang et al., 2019) | 94.5±0.3 | 98.4±0.1 | 100.0±.0 | 93.5±0.2 | 74.6±0.3 | 72.2±0.1 | 88.9 |
| ours* | 91.4±0.5 | 99.2±0.1 | 100.0±.0 | 94.6±0.5 | 76.2±0.2 | 78.0±0.2 | 89.9 |

## A.4 Additional Experiment on Office-31 Dataset

Office-31 (Saenko et al., 2010) (Fig. 9) is a popular dataset to verify the effectiveness of a domain adaptation algorithm, which contains three diverse domains, Amazon from Amazon website, Webcam by web camera and DSLR by digital SLR camera with 4,652 images in 31 unbalanced classes. In this experiment, following the protocol from Zhang et al. (2019), we evaluate our method by fine-tuning a ResNet-50 (He et al., 2015) model pretrained on ImageNet (Deng et al., 2009). The model used here is almost identical to the one in Office-Home experiment except a different width 2048 for classifiers. For optimization, we use the SGD with the Nesterov momentum term fixed to 0.9, where the batch size is 32 and learning rate is adjusted according to Ganin et al. (2016).

The results on Office-31 are reported in Tab. 4. As for the tasks D→A and W→A, judging from the adaptation accuracy of those previous methods that do not consider the joint error, it is quite likely that samples from different classes are mixed together when matching distributions. Our method shows an advantage in such case which demonstrates that the proposal manage to penalize the undesired matching between source and target. As for the tasks A→W and A→D, our proposal shows relatively high variance and poor performance especially in A→W. One possible reason is that our method depends on building reliable classifiers for the source domain to satisfy the constraint. However, the Amazon dataset contains a lot of noise (Fig. 10) such that the decision boundary of the source classifiers varies drastically in each iteration during training procedure, which can definitely harm the convergence.

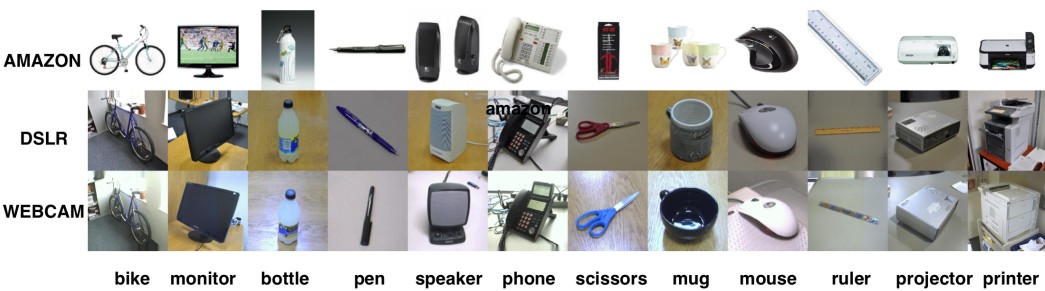

Figure 9: Sample images from the Office-31 dataset.

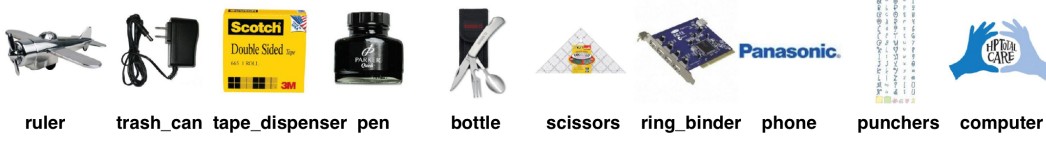

ruler     trash_can   tape_dispenser   pen     bottle     scissors   ring_binder   phone     punchers   computer

Figure 10: Noisy images from the Office-31 dataset.

