# OpenReview forum: "A General Upper Bound for Unsupervised Domain Adaptation"
_ICLR.cc/2020/Conference — Reject_

### Official Review · AnonReviewer1 · 2019-10-14
**Official Blind Review #1**

**Rating:** 1

**Review:**

This paper introduces a new upper bound of unsupervised domain adaptation, which takes the adaptability term lambda into consideration. The new theory can be expanded into a novel algorithm. Experiments on domain adaptation datasets demonstrate improvement over previous state-of-the-art methods.
The authors propose to incorporate lambda into adversarial feature learning. Specifically, the authors assume that f_s and f_t are from some hypothesis space H. Then relaxing f_s and f_t to f_1 and f_2, we can turn the problem into a minimax game between f_1, f_2 and feature extractor g. To further implement their method, the authors propose to constrain f_1 and f_2 with source accuracy and target pseudo label accuracy. Based on the margin theory, the authors also introduce the cross margin discrepancy, which increase the reliability of adversarial adaptation.
The paper is well-written and the contributions are stated clearly. The attempt to incorporate lambda into feature learning is really interesting.

However, I have several concerns:
*The proposed theory of equation (4), (5), and (6) is problematic. h is the hypothesis which belongs to a hypothesis class H. f_s and f_t are true labeling functions, and do not necessarily belong to the hypothesis space H. In this sense, the inequality of equation (4) does not hold. Problems of equation (5) and (6) are similar. The authors do realize that the supremum term can be arbitrarily large and put constraints to f_1 and f_2. But no matter what hypothesis class we are using, it generally does not contain the true labeling functions, and what we can do is only approximating them. Thus, in spite of the good performance of the proposed method, the proposed upper bound is not reliable.
*Lack of experimental results on the role of f_1 and f_2. The proposed method demonstrates good performance, but the manuscript does not provide some experimental results on the source of performance gain. In particular, how is f_1 and f_2 changed during training? Are they substantially different from the h’in MCD and MDD? Besides, how does each part contribute to the performance gain? Is it from the novel loss function or just the new adversarial adaptation method itself? A proper ablation study would be helpful.


**Experience Assessment:**

I have published one or two papers in this area.

**Review Assessment: Checking Correctness Of Derivations And Theory:**

I carefully checked the derivations and theory.

**Review Assessment: Checking Correctness Of Experiments:**

I carefully checked the experiments.

**Review Assessment: Thoroughness In Paper Reading:**

I read the paper thoroughly.

---

> ### Author Response · Authors · 2019-11-08
> **To Reviewer #1**
>
>
> Thanks for your valuable comments.
> We response to your concerns as follows.
>
> 1. We do agree that it is not easy to find the true labeling function inside a specific hypothesis space. However, we believe it is not a strong assumption that labeling functions lie in a hypothesis space with enough complexity. For instance, [1] makes a similar assumption that H includes $f_T$. Besides, since the algorithm is always run within finite samples, there is quite likely to be a function inside a specific hypothesis space that could perfectly mimic the behavior of the true labeling function on those samples. Therefore, even if the hypothesis space we use does not contain the true labeling function, it will not harm the actual training process. Moreover, the performance on 4 benchmarks (Digits, VisDA, Office-31, Office-Home) is a good evidence that our proposal is reliable when dealing with the real world problems.
>
> 2. As for the experimental results of $f_1$ and $f_2$, we've already included them inside the code we submit. The reason we did not include it in the main body of the paper is that we found it unnecessary, especially considering the page limit. The behavior of $f_1$ and $f_2$ during training process is just like h, the target accuracy growing at first then becoming stable, nothing special except that $f_2$ usually shows better performance as it leverage the information from pseudo-labeled target samples.
>
> 3. We assume you mean $f_1$ in MCD and h in MDD since h' does not appear in our paper. Actually, we do not understand why you have such concern on the difference among the obtained classifiers of ours and others, since a huge difference compared to MCD or MDD does not imply the superiority or inferiority. Currently, we can not provide the comparisons of each parameters in the learned network, however, judging from the result in Table 2 (visda dataset), our proposals tend to perform well in identifying truck, where MCD shows substantially lower accuracy. Therefore, we believe the obtained classifiers of ours and MCD are quite different.
>
> 4. We assume your last question is about the effectiveness of our upper bound and cross margin discrepancy respectively (Please correct us if we are wrong since we are not sure what do you mean by new adversarial adaptation method). The answers can be found in Table 1 and Table 2. The first row of our methods shows the effectiveness of the proposed upper bound, where we use $L_1$ as the discrepancy measurement and improves the performance from directly comparable MCD. The second row shows the result when replacing $L_1$ with cross margin discrepancy. The last row shows the result when leveraging the information from pseudo-labeled target samples to construct a more reliable hypothesis space for $f_2$. From the results, we can see they all help to improve the performance.
>
> Please do not hesitate to tell us if you have more concerns.
>
>
> [1] Mansour et al. Domain Adaptation: Learning Bounds and Algorithms. COLT, 2009

---

### Official Review · AnonReviewer3 · 2019-10-23
**Official Blind Review #3**

**Rating:** 3

**Review:**

The authors propose an approach based on an upper bound on target domain error for the task
of unsupervised domain adaptation (where one does not have access to
any labels in the target domain). The upper bound makes possible the
penalization of mixing samples from different classes together during
distribution matching.  Empirical results on image classification in a
few benchmarks show the significant promise of the approach.


The paper makes a good attempt at explaining the ideas/concepts, but
still remains very  unpolished (including English usage issues), and it is
hard to follow in a number of places. A few such, a sampling, are mentioned below with
some suggestions. In my opinion,
not ready for publication.

* abstract: '.. to address the problem for unsupervised domain
  adaptation.' >> 'to address a major problem facing many unsupervised
  domain adaptation techniques'.

* page 2, several rewordings in: "The reason is obvious, as marginal
  distributions being matched for source and target, it is possible
  that samples from different classes are aligned together, where the
  joint error becomes non-negligible since no hypothesis can classify
  source and target at the same time." For instance, a partial
  rewording: '.. when an attempt is made to match marginal
  distributions of source and target domains, samples from different
  classes can be mixed together'. Also, I wouldn't use "The reason is
  obvious"...

Finally, here it is a good place to refer to Figure 1.



* page 2: 'our proposal can degrade to some other methods' >> '.. can reduce
  to several other methods ..'



* in lines 1 and 2 on pg 3, the simplification from line 1 to 2,
  explain the major reasoning (perhaps in the appendix if you don't
  have space): there are number of terms added and subtracted (which
  is understandable), but hard to keep track of what simplifications
  are being carried and where the terms move (too many terms)...

* 'the following theorem holds' >> 'the following bound holds' (or
  inequality, etc.)

* Is derivation 3 from triangle inequality?  (add that explanation to
  line 3)

* hard to parse (missing pronoun): 'the above upper bound in minimized when h=f_s thus
  equivalent to ..'

* 'is capable of' >> 'is capable to do so' (and at this point, is not
  yet clear how the bound helps avoid the problems with distribution
  matching.. )

* Figure 1 (and subsequent figures): suggest put A, B and A' and B'
  for the two classes and domains, inside the circles, so it's easier to see what
  the source and target domain classes are (dotted boundary is hard to discern).


* on top of pg 4: couldn't the feature extractor make the max-player
  stronger too? ( In "... since the max-player taking two parameters
  f1 , f2 is too strong, we introduce a feature extractor g to make
  the min-player stronger.. " ..   )



**Experience Assessment:**

I have read many papers in this area.

**Review Assessment: Checking Correctness Of Derivations And Theory:**

I assessed the sensibility of the derivations and theory.

**Review Assessment: Checking Correctness Of Experiments:**

I assessed the sensibility of the experiments.

**Review Assessment: Thoroughness In Paper Reading:**

I read the paper at least twice and used my best judgement in assessing the paper.

---

> ### Author Response · Authors · 2019-11-08
> **To Reviewer #3**
>
>
> Thanks for your valuable comments.
> We also appreciate the suggestions for improving our writing and we will correct those lines you mentioned to make sure they can be easily understood.
> We response to your concerns as follows.
>
> 1. There is no specific reason for the simplification from Eq.1 to Eq.2 . We just rename the sum of several terms to a single term such that Eq.7 can be put inside a single line.
>
> 2. Yes. Eq.3 is from triangle inequality.
>
> 3. We explain this in Eq.3 and Eq.4 that our bound is minimized when $h=f_S$, and in such case our proposal is equivalent to an upper bound of optimal joint error. This means, ideally, our proposal can minimize the upper bound of optimal joint error (the overlapping area 2 and 5 in Fig. 1b), which is ignored by common methods. This kind of overlap can be caused by unsupervised distribution matching, since there is no guarantee that samples from different domains can be correctly matched according to their class labels (we don't have target label). Traditional methods can not prevent such mismatch since they ignore the joint error during  training  procedure, while ours can constantly penalize such undesired case.
>
> 4. We do agree that dotted boundary might be hard to discern and thanks for your nice suggestion. However, we've already put numbers and shadows inside the circles in Fig.1, Fig.2 and Fig.3 which are more essential to explaining our idea. Therefore, there is no more space to put A,B inside circles without overlap and dotted boundary seems to be our best choice currently. If you have any other suggestions, we are all ears. By the way, we take [1] as reference when drawing those figures which also uses dotted boundary.
>
> 5. No. The feature extractor will not make the max-player stronger. The max-player includes $f_1$ and $f_2$ which tries to maximize the objective, while the min-player includes h and g (feature extractor) which tries to minimize the objective.
>
> Please do not hesitate to tell us if you have more concerns.
>
>
> [1] Saito et al. Maximum Classifier Discrepancy for Unsupervised Domain Adaptation. CVPR 2018

---

> > ### Comment · AnonReviewer3 · 2019-11-14
> > **response to author replies 1 and 4**
> >
> > My question actually referred to the immediately preceding transformation:  the  line before line 1, which is not numbered, and then line 1, which replace equality with  inequality (sorry for the wrong line reference in my original comment).  In this transformation, it seems that a few minus terms are dropped (since they  are not negative, so upper bound holds), however, looks like the first term  E_T(h, f_T), also vanishes.  In any  case, it would be good to number all the lines of derivation ( or remove the numbers when unnecessary!  ) and explain that step.
> >
> > Regarding point 4, could make the  boundaries in the figures  (dotted or solid)  thicker (that's the case in Saito et al).

---

> > > ### Author Response · Authors · 2019-11-14
> > > **To Reviewer #3**
> > >
> > >
> > > Thanks for the reply.
> > >
> > > The transformation you referred to is simply owing to triangle inequality and we've already mentioned it in the text.
> > > (We didn't drop any minus terms.)
> > > Now we add another line to make the derivation more readable.
> > >
> > > As for the suggestion in point 4, we are on it (It's done now).
> > >
> > > Please do not hesitate to tell us if you have more concerns.

---

### Official Review · AnonReviewer2 · 2019-10-23
**Official Blind Review #2**

**Rating:** 6

**Review:**

Summary
-------
This paper presents a novel theoretical analysis for unsupervised domain adaptation by revisiting the \lambda joint error term charactering adaptability in the seminal analysis of Ben-David et al. They propose to replace it by considering discrepancy between information on constrained class hypothesis and a possible discrepancy term that related the learned model with the class A cross-margin discrepancy is proposed in the multiclass context. They extend their approach by proposing to reweight differently some errors with an expected performance of target models on source and an additional one where they distinguish the performance with respect to accuracy on pseudo-labeled target data. Their approach leads to an adversarial-based loss function to optimize which is evaluated on two visual domain adaptation tasks.

Evaluation
--------
The idea is novel and I find the discussion on the considered restriction on the hypothesis class interesting. The experimental evaluation is interesting with good results reported on the two problems considered. I think that some parts could be improved in terms of presentation, in particular The paper contains many typos that make sometimes the reading difficult.

Other comments
------------

-The comparison with other existing bounds is interesting. I think the authors should expand this by also taking into consideration the bound of Mansour et al., COLT'09 which has some links with the proposed approach (check their comparison to Ben-David's bound).
The links with this bound and the one of Ben-David could also be summarized in an appendix, I would like to see a better characterization of the cases where the proposed bound is better and worse.

-About original proposal. I am wondering if the authors could discuss the relationship between the value of \gamma and the expressiveness of the considered model. If the model is powerful enough, \gamma should certainly be large. In a context of a training with "Learning without forgetting" strategy, the performance on source could maintained in a high standard.
Note that in this context, the classic joint error of Ben-David et al. can be considered as rather small.

-About the alternative proposal: I am a bit skeptical on the ration \eta and (1-\eta) for accuracy on source and pseudo-labeled data respectively. Indeed, since the pseudo-labels are obtained from a classifier that make use a lot of source information, one may think that their performance is rather related. So using to correlated ratios would probably be more relevant here, but maybe the authors can bring some arguments against.

-In the experimental evaluation, the tuning of the different parameters, in particular \gamma and \mu, is not particularly discussed and there is probably an issue. I tend to think that these values are rather difficult to assess. A discussion on this point would be welcomed.

-I am not sure to understand the optimization problem (8), (9) and (18), in particular the term after the "s.t.": I would expect an inequality somewhere, otherwise everything can be added to the general objective function. If there is an alternate optimization scheme, this should be mentioned explicitly.

-Table 1 and Table 2: use a third identifier different from the second for the last version of your method.

-Please check the typos.


**Experience Assessment:**

I have published in this field for several years.

**Review Assessment: Checking Correctness Of Derivations And Theory:**

I assessed the sensibility of the derivations and theory.

**Review Assessment: Checking Correctness Of Experiments:**

I assessed the sensibility of the experiments.

**Review Assessment: Thoroughness In Paper Reading:**

I read the paper thoroughly.

---

> ### Author Response · Authors · 2019-11-08
> **To Reviewer #2**
>
>
> Thanks for your valuable and positive comments.
> We response to your concerns as follows.
>
> 1. Thanks for pointing it out and actually we have already checked Mansour et al., COLT'09. The reason we did not include it is that we find it almost identical to one of the theorem proposed in Ben-David et al. We will briefly show the difference here.
> The bound proposed in Mansour et al. can be reached by:
> $$\epsilon_T(h,f_T) \leq \epsilon_T(h,h_T) + \epsilon_T(h_T,f_T)$$
> $$= \epsilon_T(h, h_T) - \epsilon_S(h, h_T) + \epsilon_S(h, h_T)  + \epsilon_T(h_T,f_T)$$
> $$\leq disc (S,T) + \epsilon_S(h, h_S) + \epsilon_T(h_T,f_T) + \epsilon_S(h_S, h_T)$$
>
> Analogously:
> $$\epsilon_T(h,f_T) \leq \epsilon_T(h,h_S) + \epsilon_T(h_S,f_T)$$
> $$= \epsilon_T(h, h_S) - \epsilon_S(h, h_S) + \epsilon_S(h, h_S)  + \epsilon_T(h_S,f_T)$$
> $$\leq disc (S,T) + \epsilon_S(h, h_S)  + \epsilon_T(h_T,f_T)+\epsilon_T(h_S,h_T)$$
>
> This can be summarized as:
> $$\epsilon_T(h,f_T) \leq disc (S,T) + \epsilon_S(h, h_S)  + \epsilon_T(h_T,f_T)+\min (\epsilon_S(h_S, h_T), \epsilon_T(h_S,h_T))$$
> where $h_S=\mathop{\rm arg~min}_{h \in H}\epsilon_S(h,f_S)$ and $h_T=\mathop{\rm arg~min}_{h \in H}\epsilon_T(h,f_T)$
>
> While the Theorem 2 in Ben-David et al. can be expressed as:
> $$\epsilon_T(h,f_T) \leq disc (S,T) + \epsilon_S(h, f_S)  +\min (\epsilon_S(f_S, f_T), \epsilon_T(f_S,f_T))$$
>
> If we further assume H contains $f_S$ and $f_T$, these two become equivalent.
>
> Mansour et al. compared the two bounds by assuming $h_S=h_T$ , which is actually as non-realistic as  assuming $f_S=f_T$. However, we do appreciate their contributions on extending the H-divergence to general loss functions which leads to the discrepancy distance. Since our bounds does not contain the terms used in theirs, a direct comparison is a little difficult. However, [1] mentioned that MCD exploited a tighter bound of the one proposed in Ben-David et al. where the supremum term (H-divergence) is, up to a constant, equivalent to the maximal accuracy of a binary domain discriminator. And judging from the relationship between ours and MCD showed in section 3.4.2,  by choosing appropriate constraint for the hypothesis space, our bound shows superiority (at least not worse than MCD by setting $\gamma=1$). Besides, several experiments (Digit, VisDA, Office-31, Office-Home) show that our bound performs much better than those methods purely based on Ben-David et al., like ADDA and DANN.
>
> 2. We do agree that when domain shift is relatively small and the performance on source could maintained in a high standard, a large gamma always gives the best performance (our experiment results also reveal this). And in such case, the difference between ours and MCD lies only in the consideration on classic joint error (MCD does not consider the joint error). One may think we can ignore the joint error here, but according to [2], the joint error can change during distribution matching. Even if the joint error is small at the initial stage, it can grow during the training procedure. Especially when samples from different domains are mismatched, those methods which do not consider the joint error could become unbounded, while ours can constantly penalize such undesirable case.
>
> 3. We assume you suggest that we should use the correlation ratios instead of a fixed hyper-parameter $\eta$. (Please correct us if we get you wrong)
> This is indeed an interesting idea. However, if the induced feature space is well separated according to the class labels for both domains, the correlation ratios should be close to 1 (we are not so sure about it since we never check the correlation ratios). This implies the situation that source and target (solid boundary orange circle and dotted boundary orange circle in Fig. 2b) lie in the same side of $f_2$, which can not provide reliable gradient for feature extractor to bring them together. While in our proposal, by choosing an appropriate $f_2$ that could partially classify source and target (Fig. 2b), we can minimize the shadow area w.r.t the feature extractor to bring the source and target closer.

---

> > ### Author Response · Authors · 2019-11-08
> > **To Reviewer #2**
> >
> >
> > 4. We assume you mean $\gamma$ and $\eta$ since we never use $\mu$ in the paper. From Fig. 4a, Fig. 5c and Fig.5d, we can see a good performance heavily relies on the choice of hyper-parameters. However, currently we have not built an automatic approach to find the optimum, since they are closely related to the domain shift between source and target, which is usually hard to assess without using target labels (the domain shift here does not simply mean the dissimilarity of the two distributions, but the generalization performance of the feature extractor trained on source). During serval experiments, we find that setting $\eta=0.9$ works for most of the cases. We intend to tackle this challenging problem in the future work by maybe introducing some prior distributions over these hyper-parameters and optimizing the entire objective in a bayesian framework.
> >
> > 5. We use 's.t.' to emphasize that it is the constraint on the hypothesis space, where $H_1=H_{sc}$ and $H_2=H_{sc}^{\eta} \cap H_{\tilde{t}c}^{1-\eta}$.
> > We originally use something like s.t.:
> > $$\epsilon_{g(S)}(f_1)=0$$
> > $$\epsilon_{g(S)}(f_2)=1-\eta$$
> > $$\tilde{\epsilon}_{g(T)}(f_2)=\eta$$
> > However, this only holds for the 0-1 loss which can not be optimized during the training procedure. Besides, it is difficult to actually build such a constrained hypothesis space and sample from it due to a huge computational cost. Instead, a common surrogate is used to replace 0-1 loss by minimizing the weighted cross-entropy. We do agree that this part is a little confusing, and if you have any suggestions to improve the readability, we are all ears.
> >
> > 6. We use $\ast$ for the original proposal and $\star$ for the alternative proposal, so actually they are different.
> >
> > 7. Thanks for pointing it out and we will check the typos.
> >
> >
> > Please do not hesitate to tell us if you have more concerns.
> >
> >
> > [1] Kim et al. Unsupervised Visual Domain Adaptation:A Deep Max-Margin Gaussian Process Approach. CVPR,2019
> >
> > [2] Zhao et al. On Learning Invariant Representations for Domain Adaptation. ICML, 2019

---

> > > ### Comment · AnonReviewer2 · 2019-11-13
> > > **Answer 2**
> > >
> > > For point 5, I am not really convinced.
> > > I think that you could try to put some bounds (maybe with slack variables) on the quantities \epsilon_{g(S)}(f_1) and \epsilon_{g(S)}(f_2) used as constraints.

---

> > > > ### Author Response · Authors · 2019-11-14
> > > > **To Answer 2**
> > > >
> > > >
> > > > Do you mean you prefer a constraint like $\epsilon_{g(S)}(f_2)=1-\eta$ ?
> > > > If this is the case, we've explained that such constraint only holds when $\epsilon$ is 0-1 loss.
> > > > When it comes to cross entropy loss, a constraint on its quantity becomes meaningless. (At least, we are not aware of any deterministic relations between the cross entropy loss and classification accuracy.)

---

> > ### Comment · AnonReviewer2 · 2019-11-13
> > **Answer 1**
> >
> > Thank you for your answer.
> >
> > For Thm2 of Ben-David, I think a '+' is missing in the min.
> > However, my point was more to propose to expand the theoretical study by adding more ways to analyse the "quality" of the proposed bound (in the spirit of Mansour et al. approach).
> >
> > Ok for the rest.

---

> > > ### Author Response · Authors · 2019-11-14
> > > **To Answer 1**
> > >
> > >
> > > Thanks for the reply.
> > >
> > > We confirm that there is no "+" in the min term.
> > > This can be found in "A theory of learning from different domains" Mach Learn (2010) 79:page 155.
> > > (We meant to quote the Theorem 1 but we made a typo. Apologies for the confusion caused by this.)

---

### Decision · Program_Chairs · 2019-12-19

**Decision:**

Reject

**Comment:**

Given two distributions, source and target, the paper presents an upper bound on the target risk of a classifier in terms of its source risk and other terms comparing the risk under the source/target input distribution and target/source labeling function. In the end, the bound is shown to be minimized by the true labeling function for the source, and at this minimum, the value of the bound is shown to also control the "joint error", i.e., the best achievable risk on both target and source by a single classifier.

The point of the analysis is to go beyond the target risk bound presented by Ben-David et al. 2010 that is in terms of the discrepancy between the source and target and the performance of the source labeling function on the target or vice versa, whichever is smaller. Apparently, concrete domain adaptation methods "based on" the Ben-David et al. bound do not end up controlling the joint error. After various heuristic arguments, the authors develop an algorithm for unsupervised domain adaptation based on their bound in terms of a two-player game.

Only one reviewer ended up engaging with the authors in a nontrivial way. This review also argued for (weak) acceptance. Another reviewer mostly raised minor issues about grammar/style and got confused by the derivation of the "general" bound, which I've checked is ok. The third reviewer raised some issues around the realizability assumption and also asked for better understanding as to what aspects of the new proposal are responsible for the improved performance, e.g., via an ablation study.

I'm sympathetic to reviewer 1, even though I wish they had engaged with the rebuttal. I don't believe the revision included any ablation study. I think this would improve the paper. I don't think the issues raised by reviewer 3 rise to the level of rejection, especially since their main technical concern is due to their own confusion. Reviewer 2 argues for weak acceptance. However, if there was support for this paper, it wasn't enough for reviewers to engage with each other, despite my encouragement, which was disappointing.